# Controlled Thermal Release of L-Menthol with Cellulose-Acetate-Fiber-Shelled Metal-Organic Framework

**DOI:** 10.3390/molecules27186013

**Published:** 2022-09-15

**Authors:** Xinjiao Cui, Donghao Ye, Jiankun Wei, Xiaodi Du, Pengzhao Wang, Junsheng Li

**Affiliations:** 1School of Chemistry, Chemical Engineering and Life Sciences, Wuhan University of Technology, Wuhan 430070, China; 2Wuhan Marine Electric Propulsion Research Institute, Wuhan 430064, China; 3College of Chemical Engineering, Fuzhou University, Fuzhou 350108, China

**Keywords:** composite materials, stable storage, controlled release, adsorption, L-menthol

## Abstract

Fragrances have been widely used in many customer products to improve the sensory quality and cover flavor defects. The key to the successful application of fragrance is to realize controlled fragrance release, which relies on the use of an appropriate carrier for fragrance. An ideal fragrance carrier helps to achieve the stable storage and controlled release of fragrance. In this work, a novel composite fragrance carrier with MIL-101 (Cr) as the fragrance host and cellulose acetate fiber (CAF) as the protective shell was developed. The encapsulation effect of MIL-101 (Cr) and the protective function of the CAF shell significantly improved the storage stability of L-menthol (LM). Only 5 wt % of LM was lost after 40 days of storage at room temperature. Encapsulated LM could also be effectively released upon heating due to the thermal responsiveness of CAF. In addition, the composite carrier was highly stable with neglectable Cr leaching under different conditions. The results of this work showed that the developed composite carrier could be a promising carrier for the thermally triggered release of fragrance.

## 1. Introduction

Flavors and fragrances are widely used in a variety of applications such as daily necessities [1], food [2], cosmetics [3] and tobacco [4] because of their unique function to endow products with a specific sensory quality. However, most fragrances are highly volatile compounds that are easily lost, resulting in a rapid decrease in product persistence [5]. The encapsulation of fragrance through chemicals is a commonly used approach to solve this problem. Such encapsulation can be realized via the chemical binding of fragrance with the carrier. For example, β-cyclodextrin or glycoside is used to form a complex with fragrance for a sustained release [6,7]. However, the encapsulation strategy based on the chemical binding process relies on the interaction between specific functional groups in the carrier and fragrance, which limits the use of such strategy in certain fragrances with peculiar functional groups. Additionally, the complexation process may introduce organic solvents that are difficult to be completely removed. Physically encapsulation is a more feasible and generally applicable method for the controlled release of fragrance. Many physical encapsulation technologies, including freeze-drying, condensation and microfluidic technology [8], have been developed. However, the structural stability of encapsulation systems is limited. In recent years, adsorption technology is favored for practical applications because of its low cost and facile implementation process. In the adsorption-based encapsulation process, the fragrance is adsorbed into porous hosts such as zeolite [9], activated carbon [10], silica nanoparticles [11], silicoaluminate [12] and calcium carbonate [13]. Despite the fact that porous host materials could prolong the release period of the fragrance to some extent, most host materials are susceptible to environmental interference or even degradation during the fragrance adsorption process [8]. Therefore, the development of a stable host for fragrance adsorption is critical for the realization of cost-effective controlled fragrance release.

Metal-organic frameworks (MOFs) are porous crystalline materials with a periodic network structure formed by the self-assembly of transition metal ions and organic ligands and have been widely employed in adsorption-related fields, such as hydrogen storage and the selective absorption of carbon dioxide and methane from air [14,15,16], due to their high stability, flexible structure and inherent affinity toward specific molecules. MOFs have also been used to remove heavy metals and dyes from water [17,18]. In addition, some MOFs, such as the MIL series, exhibit excellent acid and alkali resistance and high thermal stability [19,20]. The chemical stability of MILs is associated with the inertness of the central metal atom, while their thermal stability is related to the strength of the metal bond [20,21,22]. Furthermore, MILs normally have a large specific surface area, which makes them excellent candidates for adsorption-related applications. In this work, we designed a composite fragrance host based on cellulose acetate fiber membrane (CAF)-coated MIL-101 (Cr). L-menthol, a widely used fragrance, was selected as a model fragrance for the evaluation of the adsorption/release performance of the host.

## 2. Results

### 2.1. Physicochemical Characterization Analysis

The design and synthesis of the fragrance release system is shown in Figure 1. MIL-101 (Cr) was chosen in this study due to its large specific surface area and high stability. To further protect the encapsulated LM from undesired loss during storage, LM loaded in MIL-101 (Cr) (LM@MIL-101 (Cr)) was coated with cellulose acetate fiber (CAF). CAF is a highly biocompatible and odor-free substance widely used in the food industry [23]. The presence of a CAF shell could firmly lock LM inside the pores of MIL-101 (Cr) under ambient conditions. Upon heating, CAF melted rapidly, and LM could diffuse from the pores of MIL-101 (Cr).

The successful synthesis of MIL-101 (Cr) was verified with XRD measurements (Figure 1A). The XRD spectra of MIL-101 (Cr) remained unchanged after the adsorption of LM, showing that MIL-101 (Cr) had a rigid structure. SEM also confirmed that MIL-101 (Cr) maintained its characteristic octahedral morphology upon LM encapsulation (Figure 1F). MIL-101(Cr) had a high LM adsorption capability due to its homogeneous size and good crystal structure. The FT-IR spectrum of MIL-101 (Cr) is shown in Figure 1B. The shifts in the corresponding IR peaks of MIL-101 (Cr) and LM@ MIL-101 (Cr) are listed in Appendix A. The alkyl characteristic peaks of LM at 2960 cm^−1^, 2930 cm^−1^ and 2870 cm^−1^ were observed in the spectra of LM@MIL-101 (Cr), indicating that LM had been successfully loaded on MIL-101 (Cr). The band width of -OH of LM@MIL-101 (Cr) significantly increased, and the band shifted toward low wavenumbers, indicating the hydrogen bonding between the OH- of LM and the Cr (OH-) of MIL-101 (Cr), which was beneficial for the adsorption of LM in MIL-101 (Cr). In addition, the peak intensity of OH- in the spectra of LM@MIL-101 (Cr) also decreased, which was also a sign of the interaction between LM and MIL-101 (Cr).

To further demonstrate that LM was infused into the pores of MIL-101 (Cr), N_2_ adsorption/desorption characterizations were conducted. The MIL-101 (Cr) isotherm was a typical type I isotherm, suggesting that the material was mostly microporous. For MIL-101 (Cr), adsorption sharply increased at low relative pressures because of the enhanced interaction between N_2_ and MIL-101 (Cr) in narrow micropores. For LM@MIL-101 (Cr) the micropore filling amount of N_2_ in LM@MIL-101 (Cr) was reduced in the low-pressure region, which was the result of the partial occupancy of LM in the micropores of LM@MIL-101 (Cr). The presence of hysteresis in the isotherm of MIL-101(Cr) may have been due to the formation of inter-particle pores due to the stacking of nano-sized MOF particles. The fitting parameters of the BET model for MIL-101 (Cr) and LM@MIL-101 (Cr) are listed in Appendix A. Both the BET specific surface area and the micropore volume of LM@MIL-101 (Cr) decreased upon LM encapsulation, clearly showing that LM was adsorbed into the microporous interior of MIL-101 (Cr). The proportions of different pore types were also calculated. The micropore volume was calculated using t-plot method and is shown in Appendix A. The total pore volume of MIL-101 (Cr) was 1.39 cm^3^/g; the micropore volume was 1.06 cm^3^/g; the mesoporous volume was 0.18 cm^3^/g; and the mesoporous content was about 13%. For LM@ MIL-101 (Cr), the total pore volume was 0.74 cm^3^/g; the micropore volume was 0.56 cm^3^/g; the mesoporous volume was 0.15 cm^3^/g; and the mesoporous content was about 20%. The SEM images showed that MIL-101 (Cr) had a regular octahedral shape without needle-like crystals on its surface, indicating that H_2_BDC had been completely removed (Figure 1E).

The stabilization of a fragrance in the carrier material is crucial for the preservation of the fragrance. A density function theory analysis using a simplified structural model was conducted to validate the high affinity between LM and MIL-101 (Cr). It should be noted that a complete structural model should be established to gain comprehensive understanding in the interaction between LM and MIL-101 (Cr). However, a DFT analysis using a complete model requires excess computational resources. Herein, a simplified structural model was used to ease the analysis process. The optimized adsorption molecular configuration is shown in Figure 2. The results showed that the hydrogen atoms on the hydroxyl group of LM tended to interact with the oxygen on the phenol hydroxyl group of MIL-101 (Cr), resulting in a stable binding configuration (E_binding_ = −2.4 eV). Such an interaction also explains the change in the intensity and position of the -OH characteristic peak in the FTIR spectra. This result means that the encapsulation of LM in the pores of MIL-101 (Cr) was thermodynamically favorable and that the adsorbed LM was stable due to the favorable binding of LM with MIL-101 (Cr).

The thermal stability of LM, MIL-101 (Cr) and LM@MIL-101 (Cr) was evaluated with TG characterizations (Figure 3A). LM lost weight continuously with heating until 165 °C, indicating that the LM had completely evaporated. MIL-101 (Cr) exhibited high thermal stability, and the weight loss between room temperature and 150 °C was ~15%, which was mainly due to the removal of water molecules from MIL-101 (Cr). The weight loss was ~50% at 260–330 °C. The decomposition of organic ligands leads to the collapse of the MIL-101 (Cr) structure [24]. In the TG curve of LM@MIL-101 (Cr), the weight loss between room temperature and 150 °C was only 5%. The low water content in the composite could have been due to the fact that LM occupied a large number of adsorption spaces and sites, which led to the amount of adsorbed water to be greatly reduced. When the temperature continued to rise, weight loss continued to occur, and the weight loss between 150 and 260 °C was ~25%. The weight loss in this state resulted from the volatilization of LM. The elevated heat-release temperature of LM indicated that the pore structure of MIL-101 (Cr) had a protective effect on LM and effectively slowed down the volatilization loss of LM. It could be seen that MIL-101 (Cr) greatly improved the thermal stability of LM.

### 2.2. Adsorption Kinetics

Next, the kinetics of LM uptake by MIL-101 (Cr) were carefully analyzed. Figure 3B shows the effect of temperature on the adsorption of LM on MIL-101 (Cr). At 120–150 °C, the adsorption amount of LM on MIL-101 (Cr) increased from 326.0 mg g^−1^ to 440.5 mg g^−1^ with the increase in temperature, which may have been caused by the increase in the saturated vapor pressure of LM with the increase in temperature [25]. At 150 °C, the adsorption capacity reached its highest (440.5 mg g^−1^), which was higher than the maximum adsorption capacity of LM on SiO_2_ (400 mg g^−1^) [4]. However, beyond 150 °C, the amount of LM adsorption on MIL-101 (Cr) gradually decreased. It may have been because the temperature was a double edge for the adsorption of LM in a sealed container sword [4]. At high temperatures, the adsorbent easily desorbed LM, reducing the adsorption contribution. The adsorption kinetics of LM on MIL-101 (Cr) were studied at the optimum adsorption temperature (150 °C). Figure 4 depicts the evolution of the adsorption capacity over time. A pseudo-first-order kinetic model and a pseudo-second-order kinetic model were utilized [26,27] to fit the adsorption kinetic curves (Equations (1) and (4)):(1)$$q_{t}=q_{e}(1-\exp\left(-k_{1}t\right))$$
(2)$$q_{t}=\frac{k_{2}q_{e}^{2}t}{1+k_{2}q_{e}t}$$
where $q_{e}$ (mg g^−1^) and $q_{t\;}$(mg g^−1^) denote the equilibrium adsorption capacity and the adsorption capacity at time t, respectively; k_1_ (min^−1^) denotes the rate constants of the pseudo-first-order kinetic model; and k_2_ (g (mg. min)^−1^) denotes the rate constants of the pseudo-second-order kinetic model. Appendix A displays the kinetic parameters derived by fitting the model to the experimental data.

Figure 4 shows the results of fitting the LM adsorption kinetics on MIL-101 (Cr). The pseudo-first-order kinetic fitting curve was more applicable to experimental data, with a fitting correlation coefficient of 0.996. The adsorption behavior of LM on MIL-101 (Cr) may be better described using a pseudo-first-order kinetic model, demonstrating that the adsorption of LM on MIL-101 (Cr) was primarily physical adsorption. As shown in Appendix A, the theoretical adsorption capacity of LM on MIL-101 (Cr) was 432.1 mg g^−1^, and the adsorption rate constant was 4.46 min^−1^.

### 2.3. Heat-Release Property Analysis

CAF was coated onto LM@MIL-101 (Cr) through a solution-mediated process. CAF was selected for coating due to its high stability and excellent biocompatibility. The thermally triggered release of LM from LM@MIL-101 (Cr)-CAF with different CAF amounts was surveyed (Figure 5A). In order to protect LM from diffusion under storage conditions, LM@MIL-101 (Cr) was further coated with cellulose acetate fiber (CAF). The release performance of LM from CAF coated LM@MIL-101 (Cr) with different CAF amounts (abbreviated as LM@MIL-101 (Cr)-CAFx, where x denotes the weight of CAF used in grams) was evaluated. To test the sealing property of CAF, the pore structure of LM@MIL-101 (Cr)-CAF_0.9_ was characterized using N_2_ adsorption/desorption experiments. In Appendix A, N_2_ adsorption/desorption isotherms and pore-size distribution maps show that it contained very few pore structures. The BET surface area was only 74.06 m^2^ g^−1^. Moreover, the total pore volume was only 0.0509 cm^3^ g^−1^. Thus, it could be proved that the material surface was almost entirely cellulose acetate fiber, and LM was sealed inside the cellulose acetate fiber membrane. The heat-release temperature of 250 °C was selected for this experiment because such a temperature is relevant for many applications, such as the release of fragrance from tobacco. The amount of LM released in different systems is shown in Appendix A. Pristine LM showed the highest release amount. When LM was coated with CAF, its release amount significantly decreased. This result indicated that the CAF shell increased the resistance for LM release. It could be clearly seen that the release rate of LM@CAF_0.9_ rapidly increased after 32 s. In contrast, the release rate of LM was significantly lower for LM@MIL-101 (Cr)-CAF. In the LM@MIL-101 (Cr)-CAF_X_ series, LM@MIL-101 (Cr)-CAF_0.9_ had the highest value of 0.082 mg, which reached 74.9% of that for pristine LM under identical conditions (Appendix A). LM release was greatly reduced when the content of CAF was lower than 0.9 g. This could have been because the amount of CAF was insufficient for the complete coating of LM@MIL-101 (Cr). The quantity of LM released also decreased with the increase in CAF when the CAF content was greater than 0.9. This was understandable, because excess CAF coating posed higher diffusion resistance for LM.

### 2.4. Storage Stability Analysis

The high stability of fragrances during storage is essential for the prolonging of the shelf life of products. The storage stability of LM was evaluated at room temperature (Figure 5B). When LM was exposed to air for 40 days, 80 wt % of LM volatilized. When LM was loaded in MIL-101 (Cr) and coated with 0.9 g of CAF, it was observed that 95% of its initial mass remained after 40 days of exposure to air. These results prove that LM@MIL-101 (Cr)-CAF_0.9_ is an excellent sustained release system of LM.

### 2.5. Carrier Stability Analysis

In order to prove that the LM carrier MIL-101 (Cr) designed in this work was stable, we collected the filter after a simulating cigarette-smoking experiment, soaked it in 5% hydrochloric acid and analyzed the content of leached Cr using atomic absorption spectrometry. The results showed that the chromium content in each cigarette was about 0.64 μg, which is far lower than the amount of chromium allowed to be in the human body (50–200 μg.d^−1^). Considering that the carrier designed in this study may be used in daily necessities, cosmetics, food, and other products that may come into contact with the human body, we further soaked MIL-101 (Cr) with normal saline to detect the loss rate of chromium. The results showed that the leaching rates of Cr were 1.75, 1.77, 2.06, 2.47 and 2.54 mg L^−1^ from the 1st to the 5th day, respectively. According to the Chinese standard (GB 2706–2001), LM content in candy, beverages, ice cream and other foods is about 0.054 wt %–0.1 wt %. If LM were encapsulated in the MIL-101 (Cr) carrier for these applications, the amount of MIL-101 (Cr) carrier would be 0.12 wt %–0.22 wt %, and the maximum amount of Cr dissolved after 5 day of soaking would be 0.39–0.56 mg kg^−1^ (Figure 6A). Such leaked amounts would also be below the allowed dosage of Cr in food (0.5–1 mg kg^−1^; GB 14961-94).

To further prove the safety performance of the carrier, MIL-101 (Cr) was immersed in normal saline for 1–5 days, and its crystal structure was analyzed using XRD (Figure 6B). The results showed that MIL-101 (Cr) retained good crystallinity after soaking in normal saline for 3 days, which suggested that MIL-101 (Cr) maintained high stability in normal saline. These results fully demonstrate that the fragrance carrier designed in this work is safe.

## 3. Materials and Methods

### 3.1. Materials

Chromium nitrate nonahydrate (Cr(NO_3_)_3_.9H_2_O, 99.0%), terephthalic acid (H_2_BDC, 99.0%), N, N-Dimethyl formamide (DMF, 99.5%), anhydrous ethanol (C_2_H_5_OH, 99.7%), L-menthol (C_10_H_20_O, 99.0%), glacial acetic acid (CH_3_COOH, 99.5%) and isopropyl alcohol (C_3_H_8_O, 99.7%) were purchased from Aladdin Reagent Co. Ltd. Cellulose acetate fiber, heated non-burning cigarette and Whatman Cambridge filters were obtained from Hubei Tobacco Co. LTD.

### 3.2. Synthesis of MIL-101 (Cr)

MIL-101 (Cr) was synthesized based on published methods [28]. In short, the corresponding proportions of H_2_BDC (2.49 g, 15 mmol), Cr(NO_3_)_3_.9H_2_O (6.0 g, 15 mmol) and deionized water (60 mL) were ultrasonically processed for 30 min. They were transferred to a PTFE autoclave and allowed to react at 218 °C for 18 h. Once cooled to room temperature, the filtered product was washed with hot DMF to remove unreacted H_2_BDC and was specifically washed with hot ethanol solution to remove DMF. After vacuum-drying at 150 °C for 10 h, pure MIL-101 (Cr) was obtained.

### 3.3. Characterization

XRD data were collected using an Empyrean diffractometer with a Cu K radiation wavelength of 1.5406 Å (40 kV, 40 mA), and the scanning range was 5°–30°. A JSM-7500F/JSM-7500F scanning electron microscope was used to perform the measurements. The accelerating voltage was 20.0 kv. FT-IR spectra were obtained using a VATAR FT-IR370 spectrometer. The nitrogen adsorption–desorption curves were measured using an ASAP2460 analyzer from Micromeritics, USA. Before the measurements, the samples were activated under vacuum at 150 °C for 12 h. After activation, the samples were degassed at 200 °C for 12 h, and the tests were performed under 77 K liquid nitrogen. The thermal stability of LM, MIL-101 (Cr) and LM@ MIL-101 (Cr) was tested using an SA449F3/SA449F3 thermal analyzer under the following conditions: air atmosphere, room temperature to 500 °C, heating rate of 10 °C min^–1^.

### 3.4. Theoretical Calculations

All structural optimizations and energy calculations were performed using Materials Studio’s DMol3 module. The generalized gradient approximation (GGA) Perdew–Burke–Ernzerhof (PBE) functional was used to calculate the exchange correlation energy. DND was used in the base group. In order to accelerate the convergence rate of self-consistent field (SCF) iteration and allow orbit relaxation to be obtained during calculation, Fermi smearing was set to 5 × 10^−3^ Ha. The convergence criteria of atomic energy change, maximum stress and displacement were set as 1 × 10^−7^ Ha, 4 × 10^−3^ Ha/Å and 5 × 10^−3^ Å, respectively. After the convergence test, the orbital cutoff precision was set to 3.3 Å, and the binding energy (E_b_) calculation formula was applied as follows:(3)$$E_{b}=E_{\mathrm{total}}-E_{G}-E_{c}$$
where E_total_, E_G_ and E_C_ stand for total energy after combination and monomer energy after structural optimization, respectively.

### 3.5. Adsorption Experiments

We placed the above synthetic carrier, MIL-101 (Cr), in a vacuum-drying oven at 150 °C for 2 h to remove water. We filled an autoclave with 0.4 g of carrier and 0.2 g of LM and kept it sealed at a specific temperature for a set period of time. To make the unabsorbed LM volatilize, we opened the autoclave lid after the adsorption process and kept it at 80 °C for 15 min. We closed the autoclave and weighed it after it had cooled to room temperature. The amount of LM adsorbed by the adsorbent per unit mass was calculated using Equation (3):(4)$$q=\frac{\left(M_{2}-M_{1}\right)*1000}{M_{1}}$$
where q (mg g^−1^) is the amount adsorbed per gram of adsorbent, M_2_ (g) is the mass after adsorption and M_1_ (g) is the mass of adsorbent before adsorption.

### 3.6. Membrane Sealing of Sample

We took more than 0.1 g of LM@MIL-101 (Cr) and added 10 mL of acetic acid solution; LM@MIL-101 (Cr) was evenly ultrasonically dispersed for 5 min. Then, we added 0.7, 0.9, 1.1 and 1.3 g of CAF to the above acetic acid solution, respectively, stirring to form a uniform slurry. The slurry was transferred to the sample plate, spread out and placed in the dark for 24 h to remove bubbles. Afterwards, deionized water was added, and the solution was placed in the dark for 24 h to completely solidify the sample. The film was dried at 30 °C prior to use. The dried film samples with different contents of CAF were named LM@MIL-101 (Cr)-CAF_0.7_, LM@MIL-101 (Cr)-CAF_0.9_, LM@MIL-101 (Cr)-CAF_1.1_ and LM@MIL-101 (Cr)-CAF_1.3_, respectively. LM was coated with 0.9 g of cellulose acetate in the same way and named LM-CAF_0.9_. The thicknesses of the LM@MIL-101 (Cr)-CAF_0.7_, LM@MIL-101 (Cr)-CAF_0.9_, LM@MIL-101 (Cr)-CAF_1.1_ and LM@MIL-101 (Cr)-CAF_1.3_ films were about 280 μm, 420 μm, 690 μm and 830 μm, respectively.

### 3.7. Heat-Release Experiment

LM@MIL-101 (Cr)-CAF_0.7_, LM@MIL-101 (Cr)-CAF_0.9_, LM@MIL-101 (Cr)-CAF_1.1_, LM@MIL-101 (Cr)-CAF_1.3_ and LM-CAF_0.9_ were individually put into the support section of a heated non-burning cigarette. Moreover, the amount of LM added to each cigarette was 1 mg. A cigarette-smoking experiment was carried out at 250 °C. The released LM was captured using Whatman Cambridge filters and quantitatively analyzed using gas chromatography.

LM was extracted for 1 h at room temperature with a 20 mL isopropanol solution containing internal standard naphthalene (20 mg L^−1^). Centrifugation was used to collect the supernatant for chromatographic analysis. We kept the DB-FFAP capillary column (30 m × 0.53 mm × 0.25 m) at 80 °C for 1 min and then increased the temperature to 220 °C at a rate of 20 °C min^−1^ for 5 min. The column flow rate was 8 mL min^−1^; the inlet temperature was 220 °C; the split ratio was 2:1; the carrier gas was H_2_; and the inlet temperature was 220 °C. The volume of the sample was 1 μL. The temperature of the FID detector was set to 260 °C. LM concentrations of 2 mg L^−1^, 4 mg L^−1^, 6 mg L^−1^, 8 mg L^−1^ and 10 mg L^−1^ were used for precise quantitative analysis.

### 3.8. Storage Stability Test

LM, LM@MIL-101 (Cr) and LM@MIL-101 (Cr)-CAF_0.9_ of a certain quality were weighed and put into the glass dryer. The LM content of the three samples was 50 mg, and the mass was weighed at the same time every day to observe volatilization at room temperature.

### 3.9. Carrier Stability Test

LM@MIL-101 (Cr)-CAF_0.9_ was put into the support section of a heated non-burning cigarette. A cigarette-smoking experiment was carried out at 250 °C. Chromium from the carrier, MIL-101 (Cr), was captured using a Whatman Cambridge filter, leached with 5% hydrochloric acid and quantitatively analyzed using atomic absorption spectrometry. In addition, 1 g of MIL-101 (Cr) was added to a 100 mL physiological saline bottle and placed at room temperature. The supernatant was taken after 1, 2, 3, 4 and 5 days and filtered using a 0.22 μm filter. The amount of chromium ion elution was analyzed using atomic absorption spectrometry.

For a further structural stability test, 1 g of MIL-101 (Cr) powder was dispersed in 100 mL of normal saline and placed at room temperature for 1–5 days. Sample volumes of 20 mL were taken out on the 1st, 3rd and 5th day; the powder was collected by centrifugation and washed with water for 3 times, and the color of the powder remained unchanged. After activation, it was used for XRD measurements.

## 4. Conclusions

A novel fragrance release system based on an MIL-101 (Cr) host and a cellulose acetate fiber (CAF) shell is developed and characterized using L-menthol (LM) as the model fragrance. Due to the unique porous structure and rich hydrogen bonding sites of MIL-101 (Cr), a high LM uptake is achieved in the fragrance carrier. In addition, the combination of the high stability of MIL-101 (Cr) and the thermally responsive protective performance of CAF endow the novel fragrance carrier with excellent fragrance preserving properties under storage conditions and desired fragrance release performance upon heating. Furthermore, the carrier system is highly stable with little Cr leakage under various conditions, which makes it applicable in a range of applications. We expect that the as-developed MIL-101 (Cr)-CAF host could be a promising carrier for heat-triggered fragrance release and that it could be beneficial for the future design of advanced fragrance release systems.

## Data Availability

Data is contained within the article or Appendix A.

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
