# Peer review of "Controlled Thermal Release of L-Menthol with Cellulose-Acetate-Fiber-Shelled Metal-Organic Framework"

_molecules, 2022, doi:10.3390/molecules27186013_

Round 1

Reviewer 1 Report

Paper by X. Cui et al. presents a novel and interesting strategy to prepare the fragrance container for tobacco industry based on supramolecular L-menthol@MIL-101(Cr) composites coated and protected by cellulose fibers (LM@MIL-101(Cr)-CAF). The particular merit of this manuscript is the evaluation of the potential risk of the LM@MIL-101(Cr)-CAF composite for the human health This manuscript may be published in Molecules after addressing the following issues.

1. Please explain more in detail, how LM@MIL-101(Cr)-CAF films are formed without substrate. What is their thickness?

2. Please provide XRD patterns of the synthesized MIL-101(Cr) and LM@MIL-101(Cr) materials in the 2-4 ° 2q region to justify the formation of MIL-101 crystalline structure, no MIL-53.

3. Please discuss the differences between N2-adsorption isotherms of MIL-101(Cr) and LM@MIL-101(Cr) materials as well as a hysteresis origin on the MIL-101(Cr) isotherm.

4. What is the mesopore content in the MIL-101(Cr) and LM@MIL-101(Cr) samples?

5. The referee has not find the characterization data for the LM@MIL-101(Cr)-CAF composites in the manuscript.

Author Response

General comments: Paper by X. Cui et al. presents a novel and interesting strategy to prepare the fragrance container for tobacco industry based on supramolecular L-menthol@MIL-101(Cr) composites coated and protected by cellulose fibers (LM@MIL-101(Cr)-CAF). The particular merit of this manuscript is the evaluation of the potential risk of the LM@MIL-101(Cr)-CAF composite for the human health This manuscript may be published in Molecules after addressing the following issues.

Response: We thank the reviewer very much for the positive comments on our manuscript. We have carefully considered all the suggestions and made a point-to point response below.

Q1: Please explain more in detail, how LM@MIL-101(Cr)-CAF films are formed without substrate. What is their thickness?

A1: We thank the reviewer very much for pointing out this important issue. The formation process of LM@MIL-101(Cr)-CAF film is as follows: Firstly, LM@MIL-101(Cr) is evenly dispersed in acetic acid solution, and a certain amount of cellulose acetate fiber is then added to the acetic acid solution, which is stirred to form a uniform slurry. The slurry is transferred to the sample plate to spread out, placed in dark for 24 hours to remove bubbles. Afterwards, deionized water is added and the solution is placed in dark for 24 hours to completely solidify the sample. The film was dried at 30 ℃ prior to use. The thickness of LM@MIL-101 (Cr)-CAF0.7, LM@MIL-101 (Cr)-CAF0.9, LM@MIL-101 (Cr)-CAF1.1, and LM@MIL-101 (Cr)-CAF1.3 films are about 280 mm, 420 mm, 690 mm, 830 mm, respectively. We have added the film formation process into the experimental section of the revised manuscript.

Q2: Please provide XRD patterns of the synthesized MIL-101(Cr) and LM@MIL-101(Cr) materials in the 2-4° 2q region to justify the formation of MIL-101 crystalline structure, no MIL-53.

A2: We thank the reviewer very much for this critical question. We think that MIL-53 is not formed in our synthesis. From Fig. 1(A), it is obvious that the peak shape and location of MIL-101(Cr) and LM@MIL-101(Cr) are consistent with the spectra of the simulated MIL-101(Cr). No additional peak was observed in the spectra. In addition,

XRD spectra of MIL-53 synthesized at high temperature (MIL-53ht) and low temperature (MIL-53lt) were simulated and compared with the XRD spectra of the synthesized sample. MIL-53ht has characteristic diffraction peaks at 9.32°, 9.68° and 18.78°, and MIL-53lt has characteristic diffraction peaks at 9.28°, 12.16° and 17.3°. These peaks are all absent in the XRD spectra of MIL-101(Cr) and LM@MIL-101(Cr), suggestion that MIL-53 is not formed in our sample.

Q3: Please discuss the differences between N2-adsorption isotherms of MIL-101(Cr) and LM@MIL-101(Cr) materials as well as a hysteresis origin on the MIL-101(Cr) isotherm.

A3: We thank the reviewer for raising this question.  The N2 adsorption/desorption isotherms of both sample is type I isotherm, suggestion the microporous feature of the samples. For MIL-101 (Cr), the adsorption increases sharply at low relative pressures because of the enhanced interaction between N2 and MIL-101 (Cr) in narrow micropores. For LM@MIL-101 (Cr), as LM occupied part of the micropores of MIL-101 (Cr), the micropore filling amount of N2 in LM@MIL-101 (Cr) was reduced in the low pressure region, which is a result of partial occupancy of LM in the micropore of LM@MIL-101 (Cr). The presence of hysteresis in the isotherm of MIL-101(Cr) may be due to the formation of inter-particle pores due to the stacking of nano-sized MOF particles. Relating discussion has been added into the revised manuscript.

Q4: What is the mesopore content in the MIL-101(Cr) and LM@MIL-101(Cr) samples?

A4: Thanks for the question. According to the N2 desorption isotherm and pore size distribution, MIL-101 (Cr) and LM@MIL-101 (Cr) are typical microporous structures, and contains a small amount of mesoporous pores The micropore volume was calculated by t-plot method and added to Table 2. The total pore volume of MIL-101 (Cr) was 1.39 cm3/g, the micropore volume was 1.06 cm3/g, the mesoporous volume was 0.18 cm3/g, and the mesoporous content was about 13%. For LM@ MIL-101 (Cr), the total pore volume was 0.74 cm3/g, the micropore volume was 0.56 cm3/g, the mesoporous volume was 0.15 cm3/g, and the mesoporous content was about 20%.

Q5: The referee has not find the characterization data for the LM@MIL-101(Cr)-CAF composites in the manuscript.

A5: We thank the reviewer for this important suggestion. Considering that both LM and MIL-101(Cr) are highly stable, we did not conduct compositional analysis (XRD) of the sample. Following the reviewer’s suggestion, we performed N2 adsorption/desorption measurements to investigate the influence of CAF coating on the porous features of the sample. The results show that CAF coating well protected LM@MIL-101(Cr)-CAF from the ambient condition. The calculated BET surface area and total pore volume is only 74.06 m2/g and 0.0509 cm3/g, respectively. The result prove that the material surface is almost entirely cellulose acetate fiber and LM is sealed inside the cellulose acetate fiber membrane. These information and relating discussion are added to the revised supporting information.

Reviewer 2 Report

The manuscript reports the inclusion of a fragrance molecule in cellulose coated MIL-101(Cr) and study its release profiles. The work is interesting, and the experiments are in accordance with the objective of the paper. The manuscript is suitable for publication after the following points are addressed.

1.      What are the conditions for activating LM@MIL-101 (Cr) for N2 adsorption experiment? Does that not affect the loading of LM molecules?

2.      The DFT calculation and its results are confusing. In the actual MOF, the carboxylates are unlikely to be free for binding except on defect sites. The authors should redo the simulation with a unit having all carboxylates coordinated to the metal nodes.

3.      A following observation is about the loss of OH peak in the IR spectrum for LM@MIL-101(Cr) from only LM spectrum (Fig. 1B). Is there a possibility of LM binding to the metal SBUs? A comment in this regard would be useful for the discussion.

4.      The procedure for coating of CAF should be included explicitly in the experimental section. Also in the main text, it should be discussed before explaining the effects of coating.

Author Response

General comments: The manuscript reports the inclusion of a fragrance molecule in cellulose coated MIL-101(Cr) and study its release profiles. The work is interesting, and the experiments are in accordance with the objective of the paper. The manuscript is suitable for publication after the following points are addressed.

Response: We thank the reviewer very much for the constructive suggestions on our manuscript. We have carefully considered all the suggestions and made all necessary revisions to the manuscript.

Q1: What are the conditions for activating LM@MIL-101 (Cr) for N2 adsorption experiment? Does that not affect the loading of LM molecules?

A1: We thank the reviewer very much for pointing out this question. The nitrogen adsorption-desorption curves were measured by the ASAP2460 analyzer from Micromeritics, USA. Before measurement, the sample was activated in vacuum at 150 °C for 12 h. After activation, the samples were degassed at 200 °C for 12 h, and the test was performed under 77 K liquid nitrogen. The activation temperature was selected to be 150 ℃ because it is the optimal temperature for LM loading (Fig.3B) and sample activation at this temperature is equivalent to prolonging the adsorption time, which would not lead to the desorption of LM encapsulated in the micropores of the sample. The details for the activation process is added to the Characterization section of the manuscript

Q2: The DFT calculation and its results are confusing. In the actual MOF, the carboxylates are unlikely to be free for binding except on defect sites. The authors should redo the simulation with a unit having all carboxylates coordinated to the metal nodes.

A2: Thanks very much for this important question. Ideally, the model for DFT calculation should contain the whole MOF structure. However, the unit cell of MIL-101(Cr) is too large. DFT investigation of such a large system is complicated and deserves a separate work. We think that simplified DFT analysis with the structural units also provides qualitative information about the interaction between LM and MIL101(Cr). Therefore, we used this simplified model for the calculation. We hope the reviewer could kindly understand our point. To avoid misunderstanding, we stated the reason for the use of the structural model in the revised manuscript.

Q3: A following observation is about the loss of OH peak in the IR spectrum for LM@MIL-101(Cr) from only LM spectrum (Fig. 1B). Is there a possibility of LM binding to the metal SBUs? A comment in this regard would be useful for the discussion.

A3: We thank the reviewer very much for reminding this. The disappearance of the characteristic peaks from hydroxyl groups of LM is definitely a sigh of its interaction with other functionalities. According to the DFT analysis (Fig 2), the interaction between LM and MIL-101(Cr) occurs between the hydrogen on the hydroxyl group of LM and oxygen on the phenol hydroxyl group of MIL-101 (Cr). We have added this discussion into the revised manuscript.

Q4: The procedure for coating of CAF should be included explicitly in the experimental section. Also in the main text, it should be discussed before explaining the effects of coating.

A4: Thanks for the kind suggestion. We have explained the procedure for coating of CAF to the experimental section and discussed it in the main text before explaining the effects of coating. The procedure for coating of CAF is as follows: Firstly, LM@MIL-101(Cr) is evenly dispersed in acetic acid solution, and a certain amount of cellulose acetate fiber is then added to the acetic acid solution, which is stirred to form a uniform slurry. The slurry is transferred to the sample plate to spread out, placed in dark for 24 hours to remove bubbles. Afterwards, deionized water is added and the solution is placed in dark for 24 hours to completely solidify the sample. The film was dried at 30 ℃ prior to use.

Round 2

Reviewer 1 Report

Authors have corrected this Manuscript according all reviewer's suggestion and comments.

The revised paper may be published in the present form.